

# Metagenomics analysis of the effects of *Agaricus bisporus* mycelia on microbial diversity and CAZymes in compost

Wanqiu Chang[1,2], Weilin Feng[2], Yang Yang[3], Yingyue Shen[2], Tingting Song[2], Yu Li[1] and Weiming Cai[2]

[1] Jilin Agricultural University, Engineering Research Centre of Chinese Ministry of Education for Edible and Medicinal Fungi, Changchun, Jilin, China
[2] Zhejiang Academy of Agricultural Sciences, Institute of Horticulture, Hangzhou, Zhejiang, China
[3] Chinese Academy of Tropical Agricultural Sciences, Environment and Plant Protection Institute, Haikou, Hainan, China

## ABSTRACT

*Agaricus bisporus* growth alters the lignocellulosic composition and structure of compost. However, it is difficult to differentiate the enzyme activities of *A. bisporus* mycelia from the wider microbial community owing to the complication of completely speareting the mycelia from compost cultures. Macrogenomics analysis was employed in this study to examine the fermentation substrate of *A. bisporus* before and after mycelial growth, and the molecular mechanism of substrate utilization by *A. bisporus* mycelia was elucidated from the perspective of microbial communities and CAZymes in the substrate. The results showed that the relative abundance of *A. bisporus* mycelia increased by 77.57-fold after mycelial colonization, the laccase content was significantly increased and the lignin content was significantly decreased. Analysis of the CAZymes showed that AA10 family was extremely differentiated. Laccase-producing strains associated with AA10 family were mostly bacteria belonging to *Thermobifida* and *Thermostaphylospora*, suggesting that these bacteria may play a synergistic role in lignin decomposition along with *A. bisporus* mycelia. These findings provide preliminary evidence for the molecular mechanism of compost utilization by *A. bisporus* mycelia and offer a reference for the development and utilization of strains related to lignocellulose degradation.

# INTRODUCTION

Agricultural biomass wastes comprise organic substances generated by humans during agricultural activities (*Malool, Keshavarz Moraveji & Shayegan, 2021*). As these wastes are produced in abundant quantities and pose disposal problems, there has been an increasing interest to develop efficient and safe strategies to utilize agricultural biomass waste (*Sherwood, 2020*; *Grimm & Wösten, 2018*). At present, compost is still a primary mode of organic matter degradation (*Wang et al., 2021a* and *Wang et al., 2021b*).

Corresponding authors
Yu Li, fungi966@126.com
Weiming Cai, caiwm527@126.com

Industrial-scale production of *Agaricus bisporus*, an edible mushroom with a long history of cultivation (*Baars et al., 2020*), has solved part of the problem of agricultural waste reuse to a certain extent (*De Andrade et al., 2008*). In the process of large-scale production of *A. bisporus*, agricultural wastes, such as wheat straw and chicken manure, are mainly used as raw materials for fermentation (*Roncero-Ramos & Delgado-Andrade, 2017*), which is both environmentally-friendly and economical, addressing the issue of reusing of agricultural waste to a certain extent (*Colmenares-Cruz, Sánchez & Valle-Mora, 2017*). In recent years, significant improvements in the *A. bisporus* cultivation process have been achieved, and the application of tunnel inoculation has altered the cultivation pattern and increased the mycelial growth rate. It has been reported that the localized tunnel-growth model achieved a 13.6% increase in *A. bisporus* growth rate, when compared with the cultivation house growth model (*Wang et al., 2021a*; *Wang et al., 2021b*). However, only a few studies have performed comparative investigations of inoculated and uninoculated mushroom compost. Some studies have suggested that *A. bisporus* mycelial growth produces a range of extracellular enzymes that are involved in the degradation of the lignocellulosic fraction in compost. Lignin is mainly degraded during *A. bisporus* mycelial growth stage (PIII), with an increase in guaiacyl lignin content (G-type lignin) (*Wood & Leatham, 1983*), and the lignin-degradation products have been speculated to be the substrate for subsequent growth of *A. bisporus* (*Jurak et al., 2015*; *Jurak, Kabel & Gruppen, 2014*). The decrease and changes in lignin during the mycelial growth stage can improve the digestibility of carbohydrates in the later growth phases. In a previous study, *A. bisporus* appeared to be the dominant fungal species based on visual observation of cropping beds. However, phospholipid fatty acid analysis (PLFA) conducted on mushroom compost revealed that *A. bisporus* mycelia accounted for 6.8% w/w of the mushroom compost after complete colonization, with only less than half of the mycelia being active (*McGee, 2017*).

Many studies have focused on the crucial role of bacteria and fungi in the degradation of organic compoundsand their diverse modes of action on organic matter decomposition. While bacterial growth becomes restricted owing to their enhanced propagation on the surface of organic matter that acts as the main source of nutrients, the fungal hyphae have a strong penetrating ability. About concerning *Agaricus* growth, both mycelia and fruiting body production are not only dependent on the mushroom itself, but also bacteria and other fungi in the substrate, and the microbial community dynamics can completely change at the end of the composting process (*Song et al., 2021*). It has been noted that the final secondary fermentation (PII) compost mainly comprised lignocellulosic components from wheat straw together with microbial biomass (*Martínez et al., 2008*). Pasteurization of the compost material before inoculation has been found to result in the predominance of fungal community in the substrate, with *A. bisporus* becoming the major fungal strain and its mycelia subsequently colonizing the substrate by degrading the organic material to release nutrients (*McGee, 2018*). However, little is known about the composition and activity of the wider fungal community in the compost substrate besides *A. bisporus* throughout the mushroom cultivation process. Therefore, the present study aimed to reveal the utilization of compost substrate before and after *A. bisporus* mycelial growth and compare the differences in the microbial communities and enzyme families in the compost.

Furthermore, the effects of *A. bisporus* mycelial growth on other microorganisms during large-scale cultivation of *A. bisporus* were determined to identify novel microorganisms with potential roles in lignin degradation.

## MATERIALS & METHODS

### Sample collection and DNA extraction

Commercial strain A15 of *A. bisporus* from Sylvan (USA) was used in this study and stored in the Engineering Research Center of Chinese Ministry of Education for Edible and Medicinal Fungi (ERCCMEEMF) at Jilin Agricultural University (Changchun, China). The compost fermentation and mycelial culture experiments were performed at Zhejiang Longchen Modern Agricultural Science and Technology Co. Ltd, Jiaxing City, Zhejiang Province, China. The compost comprised wheat straw (90 t), chicken manure (83 t), peanut meal (3 t), and gypsum (9 t). Peanut meal was added as an auxiliary nitrogen source because the components of wheat straw and chicken manure in China were different from those in Europe and America (*Jun et al., 2021*). Before commencing compost fermentation, the initial C/N ratio of the compost was adjusted to 25:1. The straws were completely dampened and piled up for 1–3 days, and then the other materials were mixed and piled again for 2–3 days. The pre-compost was placed into the tunnel for primary fermentation (PI) (*Straatsma et al., 2000*; *Mouthier et al., 2017*). The pile was turned three times on days 2, 4, and 7, respectively. The treatment parameters were adjusted based on compost temperature to allow the material temperature to reach 70 °C–80 °C and remain constant for 6 days (PI). Immediately after that, secondary fermentation (Compost-PII) was conducted for 6–7 days by pasteurization (*Vieira & Pecchia, 2018*). After secondary fermentation, inoculation was performed when the temperature decreased to 24 °C and $NH_3$ level was ≤10 mg/L (*Sharma, Lyons & Chambers, 2005*). The temperature, humidity, air pressure, air volume, and other environmental factors were adjusted using Dutch Christiaens Group equipment for the intelligent control system. Under optimal environmental conditions, *A. bisporus* mycelia could grow all over the compost after 18 days (Mycelium-PIII) (*Iiyama, Stone & Macauley, 1994*).

Subsequently, samples were collected from the uninoculated compost and mycelia-filled compost, respectively. Before sample collection, the composts in the top, middle, and bottom layers of the reactor were fully mixed (*Meng et al., 2021*). All the samples were divided into two parts: one part was stored at 4 °C for physicochemical analysis and the other was frozen at −80 °C for DNA extraction. The genomic DNA was extracted from the samples using an Omega EZNA soil DNA kit, and the integrity, purity, and concentration of the extracted genomic DNA were examined by 1% agarose gel electrophoresis (100 V, 1.5 h), NanoDrop 2000, and Qubit 3.0, respectively. The extracted DNA was stored in an ultra-low-temperature freezer at −20 °C and transported to OE Biomedical Technology (Shanghai, China) for sequencing.

### Analysis of compost physicochemical properties

The compost samples were dried in an oven at 105 °C for 5 h to assess the moisture content. The dried samples were crushed and placed in the control box of a resistance furnace at

600 °C for 2 h to determine the ash content. The total nitrogen (Total-N) and carbon (Total-C) content in the samples were determined using the Kjeldahl method and $K_2Cr_2O_3$ oxidation. Determination of the pH values with an electronic pH meter (Mettler-Toledo Instruments Co., Ltd., Shanghai, China) using 10%(w/v) sample suspensions were used. Laccase and xylanase activities in the samples were evaluated using a Solarbio kit, and lignin, cellulose, and hemicellulose components were determined according to Van Soes method.

## Metagenome sequencing, assembly, and annotation

Metagenome sequencing was accomplished using the Illumina HiSeq platform with a 500 bp sequencing library. The raw data (raw reads) quality was pre-processed using Trimmomatic (*Bolger, Lohse & Usadel, 2014*) software, and optimized sequences were spliced and assembled using MEGAHIT (*Li et al., 2015*; *Li et al., 2016*) software based on De-Bruijn graph principle, and contigs with length <500 bp were filtered out for subsequent analysis. The open reading frame (ORF) of the spliced contigs was predicted with Prodigal software (*Hyatt et al., 2010*). CD-HIT software was adopted to remove redundant and non-redundant initial unigenes. The clustering parameters included 95% identity and 90% coverage. The clean reads of each sample were aligned to the non-redundant genes set (95% identity) using bowtie 2 software to calculate the gene abundance in the corresponding samples. The representative sequences in the non-redundant unigenes set were annotated to the obtained species information according to the best alignment attained by BLASTP (E value<1e−5) to National Center for Bio-technology Information (NCBI) Non-Redundant Database (Nr). Then, the sum of gene abundances for the corresponding species was used to calculate species abundance.

## Identification of carbohydrate-active enzymes

To evaluate the carbon utilization potential of microbial communities during *A. bisporus* mycelial growth, the non-redundant genes were compared with the carbohydrate-active enzymes database (CAZy) using DIAMOND software (e<1e−5) (*Buchfink, Xie & Huson, 2015*). First, all proteins with the highest sequence similarity were screened and subjected to CAZy to search against sequence libraries with the families of glycoside hydrolases (GHs), auxiliary activities (AAs), carbohydrate-binding modules (CBMs), glycosyltransferases (GTs), polysaccharide lyases (PLs), and carbohydrate esterases (CEs). Then, the differences in the CAZyme family between the two samples (uninoculated compost and mycelia-filled compost) were compared and analyzed (*Donhauser et al., 2021*).

## Data and statistical analyses

Raw data were entered and stored in Excel. The differences among the samples were examined by independent samples $t$-tests with statistical significance at $p < 0.05$ and $p < 0.01$. The data are presented as mean ± standard deviation (SD). GraphPad Prism 8.0 software and Origin 2021 were applied for statistical analysis and plotting, and a cloud platform (http://www.cloudtutu.com/) was employed for plotting.
**Table 1  Physicochemical properties of Compost-PII and Mycelium-PIII.**

| Phase | | Total carbohydrates (w/w%) | Total nitrogen (w/w%) | Moisture(%) | Ash(%) | pH |
|---|---|---|---|---|---|---|
| Compost-II | T1 | 28.48 ± 0.20 | 2.12 ± 0.03 | 66.46 ± 0.60 | 33.6 ± 1.69 | 7.85 ± 0.09 |
| | T2 | 28.63 ± 0.18 | 2.14 ± 0.07 | 66.11 ± 0.25 | 33.54 ± 1.06 | 7.65 ± 0.25 |
| | T3 | 28.58 ± 0.06 | 2.18 ± 0.08 | 66.95 ± 1.74 | 33.96 ± 0.79 | 7.90 ± 0.20 |
| Mycelium-III | T1 | 23.59 ± 0.22 | 2.05 ± 0.02 | 59.50 ± 4.14 | 36.00 ± 1.22 | 6.30 ± 0.01 |
| | T2 | 23.74 ± 0.15 | 2.06 ± 0.02 | 61.80 ± 0.86 | 36.20 ± 1.36 | 6.24 ± 0.02 |
| | T3 | 23.44 ± 0.28 | 2.13 ± 0.06 | 63.00 ± 1.47 | 34.10 ± 2.61 | 6.30 ± 0.04 |

**Notes.**

Notes. T1,T2,T3: different trials; Results represent mean ± standard deviation ($n = 3$).

# RESULTS

## Physicochemical properties of Compost-PII and Mycelium-PIII

After completion of PII, the material temperature was reduced using fans. Subsequently, *A. bisporus* was inoculated (4‰ (w/w)) and the compost substrate was filled with mycelial growth after 18 days of incubation at 22 °C–24 °C. During this period, the water content in the compost decreased from 66.48% to 61.77% with the increase in mycelial growth, whereas the ash content increased by 1.71% (Table 1). It must be noted that the water content can affect microbial activities, which in turn can influence enzyme activities. During *A. bisporus* mycelial growth, carbon consumption predominantly increased, whereas nitrogen utilization was relatively less maintained at 2.12–2.13% and no significant difference (Table 1). After mycelial growth, the pH of the compost decreased from 7.80 to 6.27. Analysis of the cellulose and lignin contents in the compost by Van Soes method revealed that the cellulose content and lignin content significantly decreased (Fig. 1). Furthermore, evaluation of the activities of several known carbon source degradation-related enzyme families indicated a moderate increase in xylanase and laccase activities after *A. bisporus* mycelial growth, three-fold and 15-fold increase respectively (Figs. 2B & 2C). The solubility of lignin in aqueous solutions was low, and the decrease in the pH of the culture material may have a significant effect on lignin solubility. In addition, the increase in protease (Fig. 2D) also promotes better development of the mycelium (*Wang et al., 2021a* and *Wang et al., 2021b*).

## Diversity of microbial communities in Compost-PII and Mycelium-PIII

The effective data volume of each sample in this experiment was 11.23–17.22 G. The N50 statistics of Contigs were distributed between 1631–2349 bp, and the number of OFR in the set of non-redundant genes was 1029162 after redundancy. The annotation rates were 89.44%, 74.62%, 42.21% and 2.57% for the non-redundant genes compared with NR, eggNOG, KEGG and CAZy databases respectively. Metagenomics analysis indicated the dominance of bacterial community in Compost-PII and Mycelium-PIII samples (93.17% and 94.27%, respectively), followed by fungi (0.25% and 0.29%, respectively), whereas the archaeal abundance remained almost unchanged. However, the abundance of viruses declined with *A. bisporus* mycelial growth (1.18% in Compost-PII and 0.13% in Mycelium-PIII). A total of 181 phyla, 157 classes, 805 families, 3460 genera, and
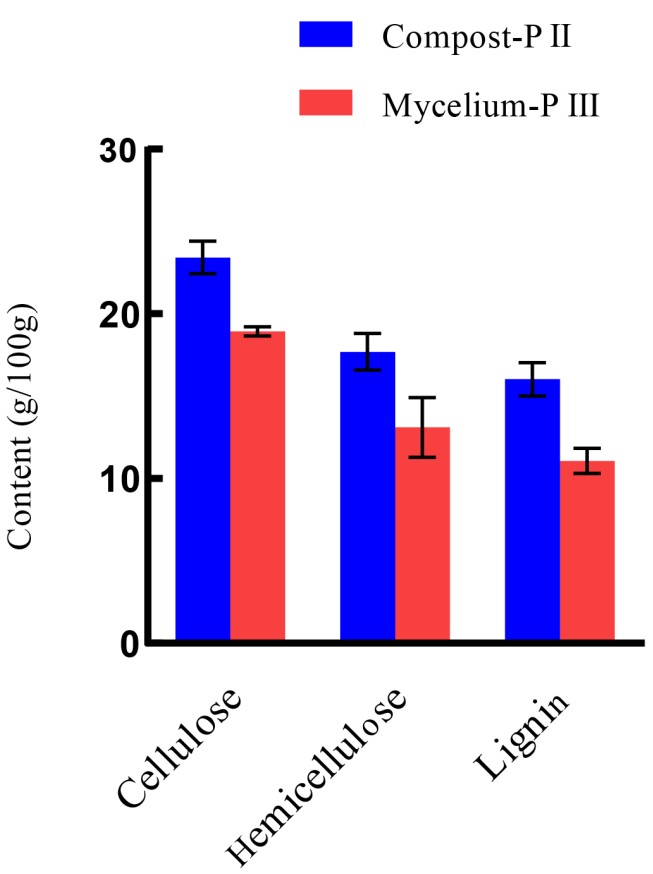

**Figure 1** **The lignocellulose contents of Compost-PII and Mycelium-PIII.** Lignocellulose: cellulose, hemicellulose, lignin. Compost-PII: end of the compost; Mycelium-PIII: *A. bisporus* mycelia could grow all over the compost after 18 days. (* $p < 0.05$, ** $p < 0.01$) ($n = 3$).

22,567 species were detected in the samples. The six most prominent bacterial phyla were Proteobacteria, Actinobacteria, Chloroflexi, Planctomycetes, Bacteroidetes, and Firmicutes (*Zhang et al., 2014*), and the abundances of Actinobacteria and Planctomycetes significantly increased after *A. bisporus* mycelial growth (Fig. 3A). Intriguingly, numerous lignocellulose-decomposing bacteria have been reported to belong to Proteobacteria, Firmicutes, Actinobacteria, and Bacteroidetes (*Lewin, Wentzel & Valla, 2013*; *Pankratov et al., 2011*). Proteobacteria and Bacteroidetes are known to play a major role in organic matter degradation and C cycling (*Wang, Mao & Li, 2018*), and Bacteroidetes can break down lignocellulose into short-chain fatty acids (*Dodd, Mackie & Cann, 2011*). It is noteworthy that although the relative abundance of the Basidiomycota phylum was low, it showed a great increase (Fig. 3B).

## Analysis of bacterial and fungal communities

In the present study, *Thermobifida, Thermostaphylospora, Sphaerobacter, Thermopolyspora, Pseudoxanthomonas,* and *Rhodothermus* were the predominant bacterial genera in the Compost-PII and Mycelium-PIII samples (*Cao et al., 2019*; *Durrant, Wood & Cain,*

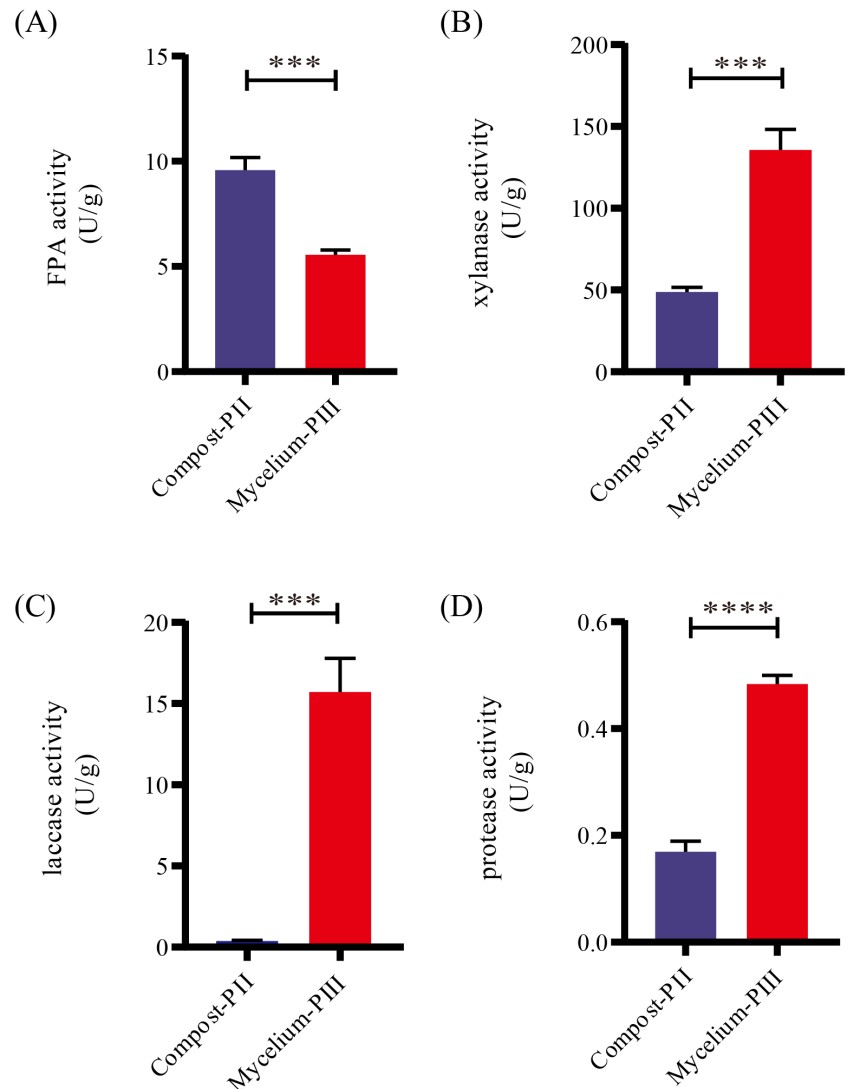

**Figure 2 Enzyme activities in various stages.** (A) FPA activity; (B) xylanase activity; (C) laccase activity; (D) protease activity. Note: * $P < 0.05$. ** $P < 0.01$. *** $P < 0.001$. **** $P < 0.0001$.

*1991*). When compared with the Compost-PII samples, the relative abundances of *Thermobifida, Thermostaphylospora, Sphaerobacter, Thermomonospora,* and *Chelatococcus* were significantly increased in Mycelium-PIII samples; in contrast, the relative abundances of *Thermopolyspora, Rhodothermus,* and *Pseudoxanthomonas* presented the opposite trend (Fig. 4A). Analyses of the microbial community composition confirmed significant shifts in the microbial community structure between the two groups, and many of the enriched genera also co-varied with function. Moreover, the variability of these microbial communities may be correlated with nutrients, compost temperature, moisture content, and pH.

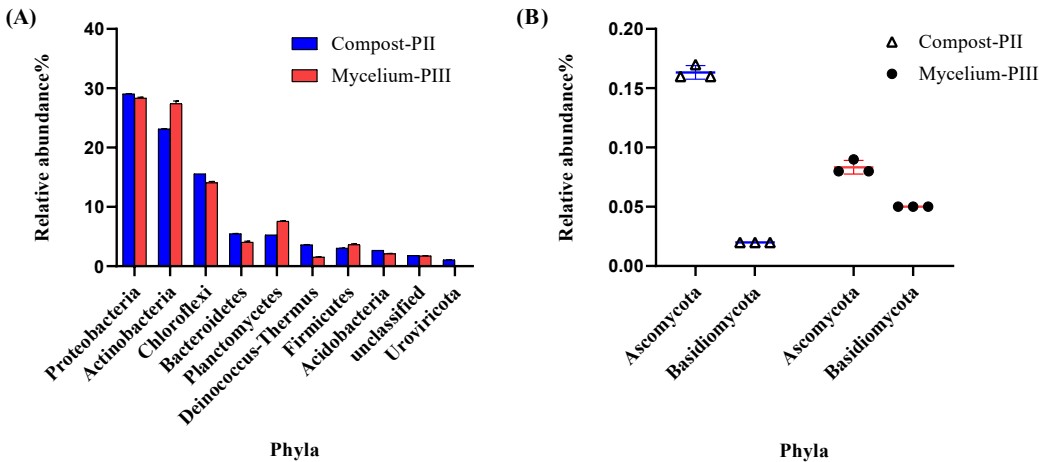

**Figure 3** **Microbial community analysis of Compost-PII and Mycelium-PIII (phylum level).** (A) Relative abundance of microbial communities (Top 10 phyla level). (B) Relative abundance of Ascomycota and Basidiomycota (phylum level).

At the species level, the relative abundance of *A. bisporus* presented the highest increase among fungi, exhibiting a 77.57-fold increase after complete mycelial growth (Fig. 4D). In addition to *A. bisporus*, the activities of other microorganisms, such as the bacteria (Fig. 4C) *Thermostaphylospora_chromogena, Thermomonospora_sp._CIF_1, Sandaracinaceae_bacterium,* and *Chelatococcus_composti,* and fungi (Fig. 4D) *Spizellomyces_punctatus, Rozella_allomycis,* and *Basidiobolus_meristosporus,* were also enhanced during *A. bisporus* mycelial growth.

## Analysis of CAZymes

The carbon-utilization potential of the microbial communities in the compost was assessed to evaluate the effects of altered substrate quantity and quality resulting from the shift in microbial activity during *A. bisporus* mycelial growth. Among the genes annotated with CAZy, in total, 431 different CAZyme families (229 GHs, 81 GTs, 48 PLs, 17 AAs, 16 CEs, and 40 CBMs) were detected in the samples. The most abundant enzyme classes at all temperatures were GHs and GTs, whereas those with the lowest abundance were PLs and AAs. At the family level, GTs were especially abundant in all the samples, with GT2 (cellulose/chitin synthase and other functions), GT4 (sucrose synthase and other functions), and GT83 (galacturonosyl transferase and other functions) being the most abundant (*Paixão et al., 2021*; *Leadbeater et al., 2021*).

Cellulose, hemicellulose, and lignin are major constituents of lignocellulose-containing raw materials (*Stech et al., 2014*). In the present study, cellulose- and hemicellulose-degrading enzymes exhibited the highest activities in Compost-PII and Mycelium-PIII samples, with GH5, GH8, and GH9 families being the predominant cellulose-degrading enzymes and GH2, GH10, GH11, GH26, and GH53 families being the major hemicellulose-degrading enzymes (Table 2). The GH2 family includes multiple enzymes, and it has been demonstrated that $\alpha$-1,3-L-arabinofuranosidase activity on substituted xylan does not
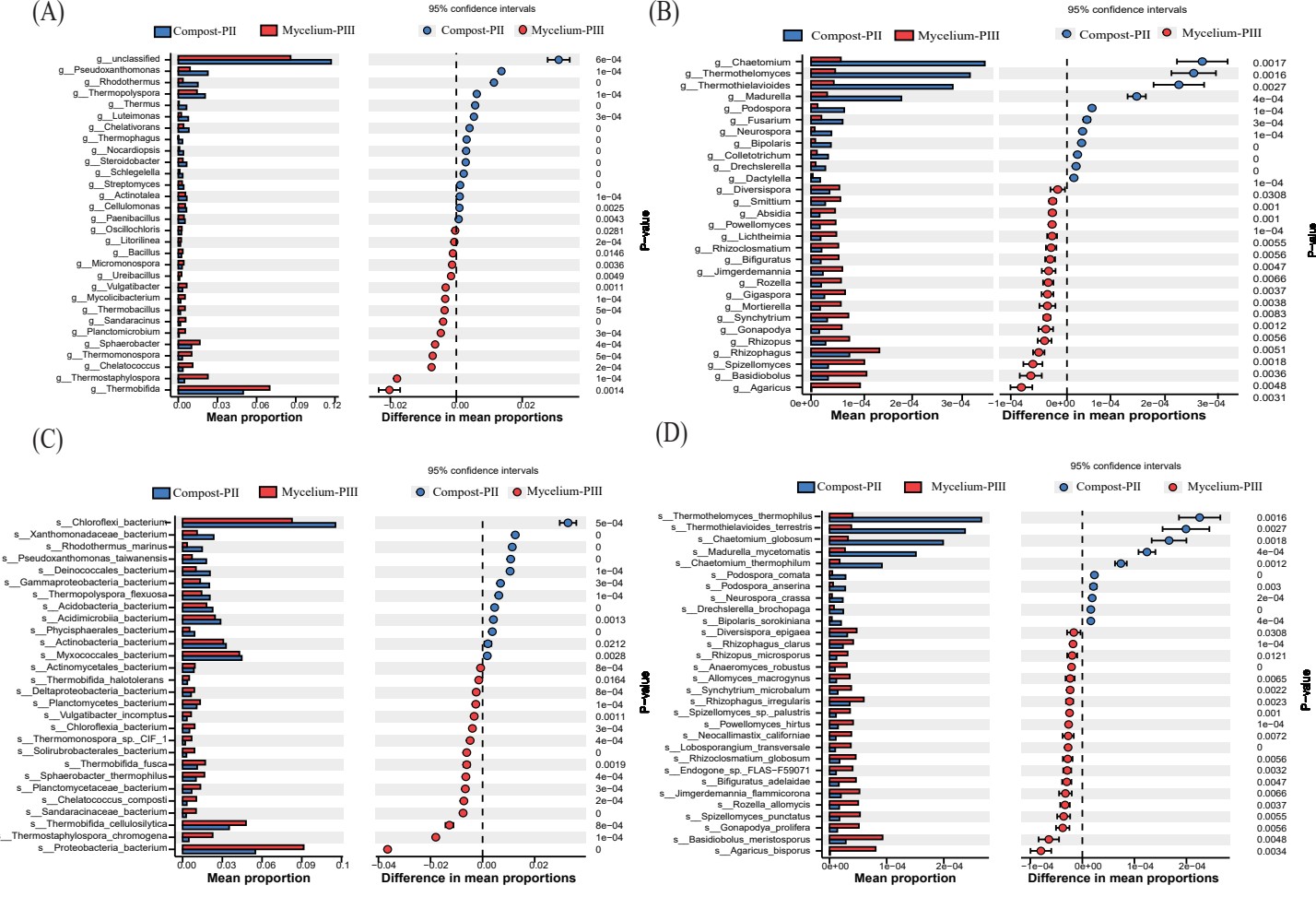

**Figure 4** **Significant differences between fungal and bacterial communities for compost-PII and mycelium-PIII on the genus level.** (A) Bacterial community; (B) fungal community and species level; (C) bacterial community; (D) fungal community.

improve compost degradation by *A. bisporus*. In nature, it is generally attributed to the metabolism of basidiomycetes white-rot fungi, since they degrade lignin more rapidly and extensively than other microorganisms (*Woiciechowski et al., 2013*). Although not a wood-rotting fungi, the *Agaricus bisporus* still plays a key role in the degradation of lignin as a grass-rotting fungi. Although AA7 family genes have been reported to play a role in lignin degradation (*Andlar et al., 2018*) and the content of these genes was relatively high in the lignin-degradation-related enzymes family in the present study, this enzymes family was not significantly different. There is a large content of the AA3 and AA6 enzyme families associated with lignin breakdown (Table 2).

## Variations in enzyme families between Compost-PII and Mycelium-PIII

The effect of *A. bisporus* mycelium on the compost substrate was mainly reflected in the enzyme families with low relative abundance at the population level. For instance, the abundances of GT32 and GH24 significantly decreased, whereas those of GH42, AA10,

**Table 2  Enzyme family identified in the metagenome.**

| CAZy family | Annotation | Genes_Count | Total putative CAZy genes, ≥50% Covered Fraction | Total putative CAZy genes, 50–70% Covered Fraction | Total putative CAZy genes, ≥90% Covered Fraction |
|---|---|---|---|---|---|
| Cellulose degrading | | | | | |
| GH5(-) | Endo-beta-1,4-glucanase ;cellulase | 73 | 50 | 19 | 0 |
| GH6(-) | Endoglucanase; cellobiohydrolase | 31 | 26 | 4 | 19 |
| GH8(↓) | chitosanase ;cellulase;licheninase | 60 | 49 | 13 | 31 |
| GH9(-) | Endoglucanase ; endo-beta-1,3(4)-glucanase | 152 | 94 | 21 | 47 |
| GH12(-) | Endoglucanase ; xyloglucan hydrolase | 38 | 36 | 0 | 33 |
| GH140(-) | Apiosidase | 41 | 24 | 8 | 9 |
| GH44(-) | Endoglucanase; xyloglucanase | 24 | 15 | 5 | 9 |
| GH116(-) | Beta-glucosidase ;beta-xylosidase | 24 | 14 | 4 | 8 |
| GH3(-) | Beta-glucosidase ; xylan 1,4-beta-xylosidase | 471 | 382 | 87 | 222 |
| GH48(-) | Endo-beta-1,4-glucanase ;chitinase | 13 | 4 | 1 | 2 |
| Hemicellulose degrading | | | | | |
| GH10(-) | Endo-1,4-beta-xylanase ;endo-1,3-beta-xylanase | 223 | 168 | 56 | 78 |
| GH11(-) | Endo-beta-1,4-xylanase ;endo-beta-1,3-xylanase | 35 | 31 | 3 | 27 |
| GH1(-) | Beta-glucosidase; beta-galactosidase | 217 | 136 | 34 | 72 |
| GH42(↑) | Beta-galactosidase; alpha-L-arabinopyranosidase | 60 | 31 | 13 | 10 |
| GH2(-) | Beta-galactosidase ; beta-mannosidase; beta-glucuronidase; alpha-L-arabinofuranosidase | 151 | 81 | 66 | 7 |
| GH26(-) | Beta-mannanase; beta-1,3-xylanase | 67 | 32 | 0 | 14 |
| GH27(↓) | Alpha-galactosidase; alpha-N-acetylgalactosaminidase | 10 | 6 | 6 | 0 |
| GH3(-) | Beta-glucosidase; xylan 1,4-beta-xylosidase; beta-glucosylceramidase | 471 | 382 | 87 | 222 |
| GH31(-) | Alpha-glucosidase; alpha-galactosidase; alpha-mannosidase; alpha-1,3-glucosidase | 164 | 102 | 36 | 54 |

**Table 2** (*continued*)

| CAZy family | Annotation | Genes_Count | Total putative CAZy genes, ≥50% Covered Fraction | Total putative CAZy genes, 50–70% Covered Fraction | Total putative CAZy genes, ≥90% Covered Fraction |
|---|---|---|---|---|---|
| GH38(-) | Alpha-mannosidase | 64 | 55 | 15 | 32 |
| GH39(-) | Alpha-L-iduronidase; beta-xylosidase | 196 | 108 | 85 | 5 |
| GH4(-) | Maltose-6-phosphate glucosidase; alpha-glucosidase | 79 | 69 | 11 | 52 |
| GH43(-) | Beta-xylosidase; alpha-L-arabinofuranosidase; xylanase | 5 | 2 | 0 | 1 |
| GH5(-) | Endo-beta-1,4-xylanase; beta-glucosidase; beta-mannosidase | 73 | 50 | 19 | 0 |
| GH53(-) | Endo-beta-1,4-galactanase | 19 | 15 | 4 | 5 |
| GH92(-) | Mannosyl-oligosaccharide alpha-1,2-mannosidase; mannosyl-oligosaccharide alpha-1,3-mannosidase | 62 | 30 | 14 | 13 |
| GH30(-) | Endo-beta-1,4-xylanase; beta-glucosidase; beta-glucuronidase | 19 | 11 | 3 | 7 |
| CE1(-) | Acetyl xylan esterase; cinnamoyl esterase | 730 | 663 | 244 | 163 |
| CE4(-) | Acetyl xylan esterase; chitin deacetylase | 623 | 589 | 87 | 251 |
| CE7(-) | Acetyl xylan esterase; cephalosporin-C deacetylase | 123 | 74 | 22 | 33 |
| CE15(-) | 4-O-methyl-glucuronoyl methylesterase | 111 | 101 | 36 | 51 |
| CE6(-) | Acetyl xylan esterase | 23 | 22 | 2 | 20 |
| GH78(-) | Alpha-L-rhamnosidase | 165 | 60 | 28 | 24 |
| GH28(-) | Polygalacturonase; alpha-L-rhamnosidase | 39 | 22 | 11 | 7 |
| GH35(-) | Beta-galactosidase; beta-1,3-galactosidase | 34 | 25 | 12 | 11 |
| PL1(↑) | Pectate lyase; pectin lyase | 9 | 9 | 0 | 2 |
| PL9(-) | Pectate lyase | 76 | 22 | 11 | 1 |
| CE8(↓) | Pectin methylesterase | 16 | 8 | 5 | 3 |
| CE12(-) | Pectin acetylesterase; acetyl xylan esterase | 20 | 20 | 1 | 16 |
| GH18(-) | Chitinase; lysozyme | 129 | 77 | 30 | 21 |
| GH19(-) | Chitinase; lysozyme | 22 | 9 | 6 | 2 |
| GH3(-) | Beta-glucosidase; xylan 1,4-beta-xylosidase; beta-glucosylceramidase | 471 | 382 | 87 | 222 |
| CE4(-) | Acetyl xylan esterase; chitin deacetylase | 623 | 589 | 87 | 251 |
| GH13(-) | Alpha-amylase; pullulanase | 77 | 59 | 15 | 28 |
| GH23(-) | Lysozyme type G; chitinase | 568 | 539 | 93 | 22 |

**Table 2** (*continued*)

| CAZy family | Annotation | Genes_Count | Total putative CAZy genes, ≥50% Covered Fraction | Total putative CAZy genes, 50–70% Covered Fraction | Total putative CAZy genes, ≥90% Covered Fraction |
|---|---|---|---|---|---|
| GH51(-) | Endoglucanase; endo-beta-1,4-xylanase; beta-xylosidase | 118 | 62 | 20 | 0 |
| GH67(-) | Alpha-glucuronidase; xylan alpha-1,2-glucuronidase | 58 | 30 | 8 | 13 |
| GH127(-) | Beta-L-arabinofuranosidase ; 3-C-carboxy-5-deoxy-L-xylose | 56 | 31 | 9 | 18 |
| GH32(-) | Endo-inulinase; | 43 | 38 | 10 | 20 |
| GH97(↓) | Glucoamylase; alpha-glucosidase; alpha-galactosidase | 41 | 21 | 9 | 11 |
| GH16(-) | Licheninase | 106 | 94 | 24 | 42 |
| GH29(-) | Alpha-L-fucosidase; alpha-1,3/1,4-L-fucosidase | 65 | 48 | 14 | 14 |
| GH95(-) | Alpha-L-fucosidase ; alpha-1,2-L-fucosidase; alpha-L-galactosidase | 53 | 30 | 13 | 13 |
| GH94(-) | Cellobiose phosphorylase ;laminaribiose phosphorylase | 54 | 29 | 15 | 2 |
| GT35(-) | Glycogen or starch phosphorylase | 162 | 92 | 69 | 11 |
| Lignin degrading | | | | | |
| AA1(↓) | Laccase/ferroxidase | 1 | 0 | 0 | 0 |
| AA2(-) | Manganese peroxidase ; versatile peroxidase ;lignin peroxidase | 39 | 12 | 1 | 7 |
| AA3(-) | Ellobiose dehydrogenase;glucose 1-oxidase | 201 | 122 | 119 | 0 |
| AA5(-) | Alactose oxidase;glyoxal oxidase ; alcohol oxidase | 8 | 0 | 0 | 0 |
| AA6(-) | 1,4-benzoquinone reductase | 98 | 86 | 10 | 61 |
| AA7(-) | Glucooligosaccharide oxidase; chitooligosaccharide oxidase | 405 | 140 | 68 | 49 |
| AA10(↑) | Lytic chitin monooxygenase | 22 | 21 | 0 | 20 |

**Notes.**
 " ↑ " and " ↓ " indicate the up-regulation and down-regulation of the relative abundance for differential CAZy, "-" indicate that there is no difference between the two comparison groups.

and CBM13 significantly increased. Furthermore, the abundance of GH8 presented a slight decrease (Fig. 5A). GH24 is known to act in association with lysozyme, GH8 is the main family of enzymes involved in cellulose degradation, and GH42 plays an important role in hemicellulose degradation. *Thermostaphylospora* belonging to Actinobacteria mainly causes an increase in the activities of $\beta$-galactosidase and $\alpha$-L-arabyranosidase of the GH42 family in the galactose metabolism pathway during the mycelial growth stage, and CBM67 has been reported to exhibit $\alpha$-L-rhamnose-binding activity (*Fujimoto et al., 2013*).

In the present study, the increase in the laccase content during the mycelial growth stage was mainly related to the AA10 family (Table 2). The major strains that cause differential laccase production are *Thermobifida, Thermostaphylospora,* and *Cellulomonas*, and in the

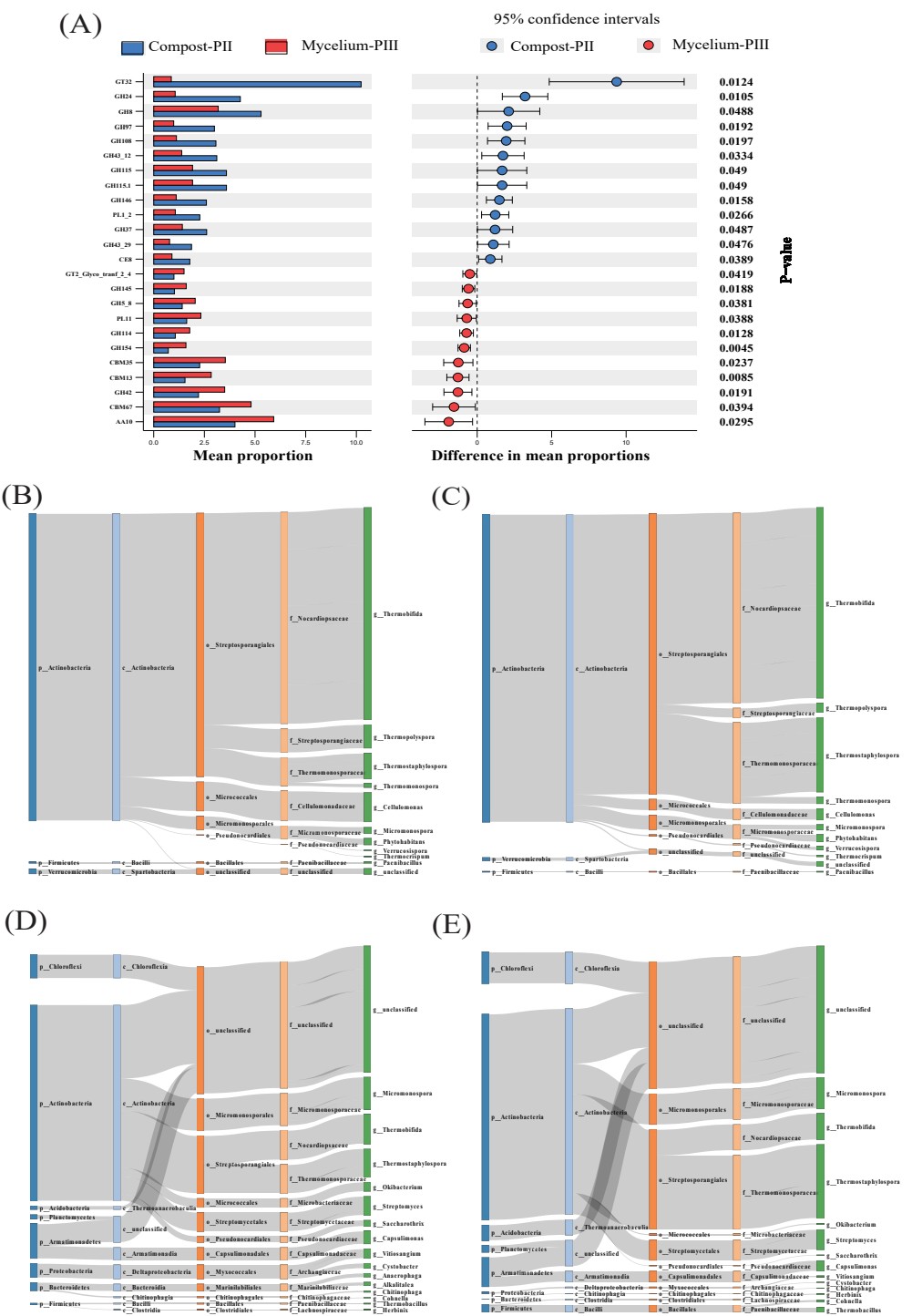

**Figure 5** **Differentiated analysis of enzyme families.** (A) Difference of putative carbohydrate-activite enzymes of two groups samples. (B) Sankey plots for AA10 in compost-PII. (C) Sankey plots for AA10 in mycelium-PIII. (D) Sankey plots for cbm13 CBM13 in compost-PII. (E) Sankey plots for CBM13 in mycelium-PIII.

present study, the relative abundances of *Thermobifida* and *Cellulomonas* decreased, while that of *Thermostaphylospora* increased (Figs. 5B & 5C). Furthermore, the increase in the xylanase content during *A. bisporus* mycelial growth stage was mainly related to the CBM13 family, and *Thermostaphylospora* was also the dominant genus that caused this increase (Figs. 5D & 5E).

## DISCUSSION

Degradation of lignin during the composting process has been reported to be caused by certain fungi and several species of bacteria and Actinomycetes (*Kabel et al., 2017*; *Fermor & Wood, 1981*; *Del Cerro et al., 2021*; *Ayuso-Fernández, Ruiz-Dueñas & Martínez, 2018*). In the present study, the artificial introduction of *A. bisporus* at the end of the composting process and the resultant dominance of *A. bisporus* in the fungal community played a major role in lignin degradation. A similar finding has also been reported by Jean-Michel Savoie et al. (*Savoie, 1998*; *Zhang et al., 2019*). During the mycelial growth of *A. bisporus,* the oxidative phosphorylation pathway became dominant, which subsequently affected the bacterial and fungal communities composition, and a part of lignin degradation originated from bacterial action. *Jurak, Kabel & Gruppen (2014)* showed that xylan solubility increased by 20% during mycelial growth, indicating partial degradation of the xylan skeleton. In the present study, xylan degradation was mainly associated with the action of *Thermostaphylospora*. However, despite xylan degradation, the carbohydrate composition and degree of substitution of xylan in the compost at the beginning and end of the mycelial growth stage were rather similar (*Jurak, Kabel & Gruppen, 2014*). The protease activity that was measured had an increasing trend. This might imply that after accessing the mycelium of *Agaricus bisporus*, the raw material chicken manure and wheat straw were used for the breakdown of the macromolecular nitrogen source material and subsequently for the development and growth of their mycelium (*Wang et al., 2021a* and *Wang et al., 2021b*).

In a previous study, some researchers (*Zhang et al., 2014*) determined the rRNA gene copy number of Bacteria and Fungi during the composting process by quantitative PCR, and found that the fungal genus *Agaricus* and unknown fungal community accounted for 45% and 55% of the microbial community, respectively, while the bacterial genus *Streptomyces* accounted for 60% of the total bacterial community during the mycelial growth phase. In contrast, in the present study, although the abundance of *Streptomyces* varied before and after the mycelial growth stage, the difference was not significant. Moreover, besides *Agaricus*, *Basidiobolus* and *Spizellomyces* were also the predominant fungi, thus providing further insights into the composition of the fungal community in the compost.

During the *A. bisporus* mycelial growth stage, the relative abundances of gene sequences still remained high at high-temperature composting, and although the abundance of fungal communities increased, the number of bacterial genes was much higher than that of fungal genes. However, as it was not possible to determine the number of active bacteria during this stage, microbial communities with a higher relative abundance of gene sequences were compared, and *A. bisporus* was found to be dominant in the fungal community. While enzymes related to lignocellulose breakdown were not detected in the macrogenome of *A.*

*bisporus*, analysis of whole-genome data of *A. bisporus* substrate confirmed the presence of a large number of genes encoding lignocellulose-degrading enzymes (*Morin et al., 2012*), because macrogenome sequencing results did not assemble genes encoding lignocellulose-degrading enzymes in *A. bisporus*. Moreover, variations were also noted in the expression of genes encoding CAZymes between compost-grown mycelia and fruiting body, with genes encoding plant cell wall degrading enzymes detected in compost-grown mycelia, but largely undetected in the fruiting body. Similarly, *Patyshakuliyeva et al. (2013)* also confirmed that compost-grown mycelia could express a large variety of CAZyme genes related to the degradation of plant biomass components. In addition, transcriptomics and proteomics investigations performed in a previous study also demonstrated that genes related to lignin degradation were only highly expressed on day 16 of mycelial growth, indicating that lignin was degraded at the initial stage of mycelial growth and was no longer altered after complete growth of mycelia (*Patyshakuliyeva et al., 2015*). Moreover, compost-grown mycelia were found to express a large number of CAZymes-encoding genes associated with the degradation of plant biomass components. In summary, the present study uncovered lignocellulose-degrading microorganisms and enzyme expressions in bacteria during *A. bisporus* mycelial growth stage in the composting process, providing further insights into lignin degradation in compost. Lignin degradation was mostly bacteria, and the main laccase-producing bacteria belonged to *Thermobifida* (*Mironov et al., 2021*; *Vanee, Brooks & Fong, 2017*) and *Thermostaphylospora*, with *Thermostaphylospora* presenting a significant increase. The results obtained can further strengthen our understanding of the specificity of *A. bisporus* mycelial growth.

## CONCLUSIONS

The present study found the laccase content was increased during the *A. bisporus* mycelial growth stage. Laccases belonging to the AA10 family were mainly derived from *Thermobifida* and *Thermostaphylospora*, the potential for lignin-degrading enzymes of bacterial origin may be grossly underestimated. *Agaricus* proliferation may require an interacting consortium of both bacteria and fungi for effective lignocellulose degradation. The results obtained offer insights into the difference in enzyme activities between *A. bisporus* mycelia and other microbial communities, and enhance our understanding of the changes in microbial communities and enzyme families during *A. bisporus* mycelial growth phase in the composting process.

### Funding

This work was supported by the China Agriculture Research System (CARS-20), the Zhejiang Science and Technology Major program on Agriculture New Variety Breeding (2021C02073), the Central Public-interest Scientific Institution Basal Research Fund (NO. 1630042022003), and the National Natural Science Foundation of China (U20A2046). The

funders had no role in study design, data collection and analysis, decision to publish, or preparation of the manuscript.

## Grant Disclosures

The following grant information was disclosed by the authors:

China Agriculture Research System: CARS-20.

Zhejiang Science and Technology Major program on Agriculture New Variety Breeding: 2021C02073.

Central Public-interest Scientific Institution Basal Research Fund: 1630042022003.

National Natural Science Foundation of China: U20A2046.

## Competing Interests

The authors declare there are no competing interests.

## Author Contributions

- Wanqiu Chang conceived and designed the experiments, performed the experiments, analyzed the data, prepared figures and tables, authored or reviewed drafts of the article, and approved the final draft.
- Weilin Feng conceived and designed the experiments, authored or reviewed drafts of the article, and approved the final draft.
- Yang Yang analyzed the data, authored or reviewed drafts of the article, and approved the final draft.
- Yingyue Shen conceived and designed the experiments, authored or reviewed drafts of the article, and approved the final draft.
- Tingting Song conceived and designed the experiments, authored or reviewed drafts of the article, and approved the final draft.
- Yu Li conceived and designed the experiments, authored or reviewed drafts of the article, and approved the final draft.
- Weiming Cai conceived and designed the experiments, analyzed the data, authored or reviewed drafts of the article, and approved the final draft.

## Data Availability

The sequence data are available in the NCBI Sequence Read Archive: PRJNA859554.

## Supplemental Information

Supplemental information for this article can be found online at http://dx.doi.org/10.7717/peerj.14426#supplemental-information.

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
