# Peer review of "Metagenomics analysis of the effects of Agaricus bisporus mycelia on microbial diversity and CAZymes in compost"

_PeerJ, doi:10.7717/peerj.14426_

## Round 0.1 · original submission · Major Revisions

The article "Metagenomics analysis of the effects of Agaricus bisporus mycelia on microbial diversity and CAZymes in compost" discusses fermented A. bisporus substrate. In the current study, A. bisporus was examined both before and after mycelial growth using macrogenomics analysis. This research helped to clarify the molecular mechanism of A. bisporus substrate utilisation. The study is fascinating, and valuable data has been generated for the benefit of a large group of researchers.

The abstract does not do a good job of presenting the information generated by the metagenomic analysis. Several sentences in the results section require a detailed explanation and statistical findings. The discussion section is tedious. There are numerous typographical errors that should be corrected during revision (See attachment). We decided that your manuscript could be reconsidered for publication based on the comments if the authors are willing to make significant revisions.

Reviewer 1 ·

Basic reporting

The fermentation substrate of A. bisporus was examined both before and after mycelial growth in the current study using macrogenomics analysis. This analysis helped to clarify the molecular mechanism of substrate utilisation by A. bisporus mycelia from the perspective of microbial communities and CAZymes in the substrate. The study is interesting and valuable information has been generated for the benefit of a large group of researchers. However, there are several issues that must be addressed for the suitability of this manuscript in the esteemed journal. The information generated through the metagenomic analysis is not very well presented in the abstract. Several sentences in the results section require a detailed explanation and findings from the statistical point of view. The discussion section is repetitive. There are plenty of typographical errors that should be rectified in revision. More specific comments are enlisted here
1. Line 22-25. The opening sentences seem incomplete and not clear in delivering the intended message.
2. The abstract requires extensive rephrasing as it lacks in providing the major scientific findings of the experiments.
3. Please give appropriate citations to the factual sentences across the manuscript. For e.g. Line 54-61: these facts require proper citations.
4. Line 104: Use of peanut meal as auxiliary nitrogen source- may be supplemented with previous references.

Experimental design

5. The material method section “Sample collection and DNA extraction should be supplemented with proper citations.
6. The information about the total replicates used in this analysis is lacking.
7. What was the actual time point when the samples were collected for further analysis?

Validity of the findings

8. The results section “Physicochemical properties of Compost and Mycelium” need a detailed explanation of the results with special emphasis on all the enzymes. The findings of rise in protease levels require further discussion.
9. Metagenomics analysis indicated the dominance of bacterial community in Compost-PII and Mycelium-PIII samples. Is there any previous report highlighting similar findings because the fungal communities were too less?
10. What could be the probable reason of virus decline due to mushroom growth?
11. Line 199-200 is it regarding bacteria only? Specify
12. Line 199-210 typographical errors must be eliminated.
13. The captions of Figures 1,2,3 may be elaborated and statistical explanations should be incorporated.
14. In Figure 4 the names of fungal and bacterial communities are not visible the font size may be increased.
15. Discussion has been introduced in the results portion also and a separate section of discussion is also provided. Please follow the journal guidelines and prepare the discussion part accordingly.

Reviewer 2 ·

Basic reporting

The language used throughout this manuscript is clear and professional. The number and quality of references seems sufficient. The Introduction does a good job of framing the work and justifying the scientific questions being addressed.

Overall, the figures are well done, but I would have liked to have seen the authors present composition at the genus level in Figure 3 to get a grasp on more specific community effects of the addition of Agaricus bisporus. Also, I do have some concerns about Figure 3 and the methods used to generate the data presented within, which are presented below. The text on Figure 4 is also quite difficult to make out, and I am not sure the taxonomy prefix (g_) is necessary - also see comments about relative abundance methods below. Similar with Table 2, its quite large and hard to really have strong take aways result-wise, perhaps make this table supplementary and have a figure that represents substantial changes? Also - for the sankey plots in Figure 5 I was expecting to see the connections between taxa and enzyme function, right now it appears to not really compliment 5A.

Experimental design

I am not really comfortable with the way the relative abundance and community analyses were performed. Relative abundance estimated using the author's odd methodology seems like it can get greatly skewed in metagenomic data, as fungi not only typically have much larger genomes, but also generally have larger genes and larger amounts of non-coding regions in their genomes. Perhaps if the authors used a read based classifier such as GOTTCHA2 or Kraken2 they would get a more accurate representation of relative abundance. It is also it a bit unexpected how the authors do not try to relate community composition to the enzyme activities/estimates - especially since in the introduction they suggest the fungal addition will cause substantial changes in both. Changes in functional potential can be a result of alterations to community structure, interactions and dynamics - but also as a result of changes to individual taxa. Since the authors went through the trouble of generating unigenes, it would be helpful to see stronger examinations of contributions of individual taxa or lineages to predicted enzyme function.

Validity of the findings

With respect to my comments above, it is hard to fully understand the impacts of A. bisporus in the context of these experiments. There are certainly changes that result from the addition, but given that fungi seem to be fairly non-dominant based on sequence recovery - its hard to understand how they may be expected to play a significant role in the processes/changes observed. Also, due to the likely inaccurate way of measuring relative abundance, it is also hard to trust the community composition analyses.

Additional comments

Overall the work presented here seems well justified and logical, but the unfortunate use of non-standard methods makes it difficult to trust the results and any conclusions drawn from them. Perhaps comparisons with more standard approaches would help increase confidence in this approach.

·

Basic reporting

The manuscript is well written. The English prolificacy need to be improved.
The research question is very valid.
The latest literature related to this study must be cited in introduction and discussion.
The figures and tables are up to the mark.

Experimental design

The experiment is well designed. I fail to understand the existence of diverse microbial community in well pasteurized compost.
Make clear, why did you compare compost-PII stage with PIII stage of mycelium?
Replication of experiment is not mentioned in physiochemical properties analysis.

Validity of the findings

The experimental results are nice validated.
Most of the data is statistically sound.
The supplementary data is provided for authenticity.
The conclusion should be short and crisp, not descriptive.

Additional comments

The discussion is mixed in results section with citations, although separate section as discussion is also available.
Thermobifida, Thermostaphylospora and Thermomonospora increase in mycellium-PIII in contrast Thermopolyspora which decreases. All are thermophilic bacteria give expected reason for this in discussion with citation of literature.

---

## Round 0.2 · accepted · Accept

The manuscript is well written. The English is improved in the revised manuscript.

All the comments are well addressed by the author. The experimental results are nicely validated. Most of the data is statistically sound. The supplementary data is provided for authenticity.

Reviewer 1 ·

Basic reporting

The queries are very well addressed and the manuscript is improved based on the suggested lines.

Experimental design

The design of the experiment is appropriate.

Validity of the findings

The findings are valuable and informative.

·

Basic reporting

The manuscript is well written. The English prolificacy is improved in revised manuscript.
All the comments are well addressed by author.

Experimental design

The experiment is well designed. My query about "Comparison of compost-PII stage with PIII stage of mycelium?" is answered by author in rebutting letter.

Validity of the findings

The experimental results are nice validated. Most of the data is statistically sound.
The supplementary data is provided for authenticity.
My concern was that "Conclusion was too descriptive". Author took it into consideration, in revised manuscript conclusion is short and crisp.

Additional comments

The authors have addressed all the comments in the revised manuscript. It should be accepted for publication.